# The Impact of the first COVID-19 shelter-in-place announcement on social distancing, difficulty in daily activities, and levels of concern in the San Francisco Bay Area: A cross-sectional social media survey

Holly Elser[1,2☯], Mathew V. Kiang [2,3☯], Esther M. John[3], Julia F. Simard[3], Melissa Bondy[3], Lorene M. Nelson[3], Wei-ting Chen[4], Eleni Linos[3,5]*

1 Stanford Medical School, Stanford University, Stanford, CA, United States of America, 2 Center for Population Health Sciences, Stanford University, Stanford, CA, United States of America, 3 Department of Epidemiology and Population Health, Stanford University, Stanford, CA, United States of America, 4 Office of Community Engagement, Stanford University, Stanford, CA, United States of America, 5 Department of Dermatology, Stanford University, Stanford, CA, United States of America

☯ These authors contributed equally to this work.
* linos@stanford.edu

**Data Availability Statement:** Original, individual-level data tied to locations, and time are considered

## Abstract

### Background

The U.S. has experienced an unprecedented number of orders to shelter in place throughout the ongoing COVID-19 pandemic. We aimed to ascertain whether social distancing; difficulty with daily activities; and levels of concern regarding COVID-19 changed after the March 16, 2020 announcement of the nation's first shelter-in-place orders (SIPO) among individuals living in the seven affected counties in the San Francisco Bay Area.

### Methods

We conducted an online, cross-sectional social media survey from March 14 –April 1, 2020. We measured changes in social distancing behavior; experienced difficulties with daily activities (i.e., access to healthcare, childcare, obtaining essential food and medications); and level of concern regarding COVID-19 after the March 16 shelter-in-place announcement in the San Francisco Bay Area versus elsewhere in the U.S.

### Results

In this non-representative sample, the percentage of respondents social distancing all of the time increased following the shelter-in-place announcement in the Bay Area (9.2%, 95% CI: 6.6, 11.9) and elsewhere in the U.S. (3.4%, 95% CI: 2.0, 5.0). Respondents also reported increased difficulty obtaining hand sanitizer, medications, and in particular respondents reported increased difficulty obtaining food in the Bay Area (13.3%, 95% CI: 10.4, 16.3) and

personally identifiable health information. These data cannot be shared owing to risks of breaching patient confidentiality, as determined by the Institutional Review Board at Stanford University. Data are available upon request (contact via irbeducation@stanford.edu) for researchers who meet the criteria for access to confidential data.

**Funding:** EL is supported by the NIH (grants DP2CA225433 and K24AR075060). MVK is supported by the National Institute on Drug Abuse (T32DA035165). LMN is supported by the Clinical and Translational Science Award Program of the National Institutes of Health's National Center for Advancing Translational Science (UL1 TR001085). The content is solely the responsibility of the authors and does not necessarily represent the official views of the NIH.

**Competing interests:** The authors have declared that no competing interests exist.

elsewhere (8.2%, 95% CI: 6.6, 9.7). We found limited evidence that level of concern regarding the COVID-19 crisis changed following the announcement.

## Conclusion

This study characterizes early changes in attitudes, behaviors, and difficulties. As states and localities implement, rollback, and reinstate shelter-in-place orders, ongoing efforts to more fully examine the social, economic, and health impacts of COVID-19, especially among vulnerable populations, are urgently needed.

## Introduction

The coronavirus disease 2019 (COVID-19) pandemic began when clusters of "pneumonia of unknown etiology" were identified in December 2019 [1–5]. By December of 2020, there were over six million confirmed cases globally. Nearly one-fourth of these confirmed cases occurred in the United States (U.S.), with over 290,000 recorded deaths to date [6,7]. In the absence of vaccines or treatments [8], the primary defense has been to reduce the risk of SARS-CoV-2 exposure through non-pharmaceutical interventions (NPIs) such as school closures, social distancing, isolation and quarantine, and use of personal masks [9–13]. NPIs were shown to be effective during the 2003 severe acute respiratory syndrome coronavirus (SARS-CoV) outbreak [14], and quickly became the cornerstone of mitigation and intervention strategies for COVID-19 globally [15–17]. However, the extent and level of enforcement of these measures vary widely [9].

On March 19, 2020, California was the first U.S. state to enact a statewide shelter-in-place order (SIPO) [18], following an announcement on March 16, 2020 of a SIPO for seven San Francisco Bay Area counties effective on 12:01 AM on March 17, 2020 [19]. In the following weeks, 42 states and the District of Columbia passed such orders [20]. Subsequent SARS-CoV-2 wintertime outbreaks may necessitate repeated intermittent social distancing orders into 2021 [17]. Given the unprecedented nature of SIPOs in the U.S. and disjointed efforts by local and state governments, school districts, and universities to enact, rollback, and re-enact SIPOs, it is critical that we understand the impact of these orders on the public's behaviors and perceptions.

For the present study, we employed convenience sampling to rapidly ascertain and summarize the impact of the announcement of the nation's first SIPO on March 16, 2020. More specifically, we use a difference-in-differences estimator to estimate changes among respondents living in the seven counties in the San Francisco Bay Area affected by the announcement versus those living elsewhere in the U.S. A large body of COVID-19 literature has employed quasi-experimental methods to examine the impact of the pandemic on a variety of topics including superspreader events [21–23], air pollution [24,25], unemployment [26], and demand for online shopping [27]. Many of these quasi-experimental studies have focused on changes in human mobility using aggregated smartphone-based measures such as time spent at home [28,29]. Relatively fewer studies have applied quasi-experimental techniques to examine the impact of SIPOs, and those that do often focus on cases, hospitalizations, mortality, or transmission [30–32].

The present study adds to the growing literature on the complex and varied impacts of SIPOs by characterizing not only the degree of behavior change (i.e. levels of social distancing), but also by characterizing how difficulty related to daily activities such as obtaining food,

essential medications and childcare and levels of concern regarding the COVID-19 crisis changed in the wake of the announcement.

## Methods

### Study sample

We conducted a cross-sectional, online survey with convenience sampling through three social media platforms (NextDoor, Twitter, and Facebook) from March 14, 2020 through April 1, 2020. Twitter and Facebook posts were shareable to facilitate snowball sampling. We included all respondents who completed at least 80% of the survey and excluded those missing both zip code and GeoIP location and those outside of the U.S.

### Data collection

The 21-item survey collected information regarding level of sheltering in place, experienced difficulty with daily activities, level of concern, demographic characteristics, and location. Demographic information included gender (female, male, other); race/ethnicity (white, Asian/ Pacific Islander, Hispanic/Latino, Black or other); year of birth was used to create age categories (25 years or less; 26–45; 46–65; older than 65 years); education (less than high school, high school or GED, some college, bachelor's degree); and health insurance (yes, no, don't know). Respondents reported the number of children (<18 years) and adults over age 65 years in their household. Participants were informed of the purpose, risks, and benefits of the study and provided their consent to participate in the survey.

### Shelter-in-place announcement

We focused the analysis on the implications of a SIPO announced for six San Francisco Bay Area counties (San Francisco, Santa Clara, San Mateo, Marin, Contra Costa, and Alameda) and separately for Santa Cruz County (hereafter, referred to collectively as "seven Bay Area counties") made mid-day on March 16, 2020. The announcement preceded the implementation of the order to shelter-in-place in the aforementioned counties on March 19, 2020 by three days. We classified survey responses collected before March 16, 2020 as having occurred before the announcement of the SIPO. We did so in order to more precisely identify responses that occurred before the announcement, as we anticipated that some respondents were aware of or suspected the announcement several hours before it occurred.

### Respondent locations

We differentiated survey respondents living in the seven affected Bay Area counties from those residing elsewhere in the U.S. using self-reported zip codes. Self-reported zip codes were mapped to Bay Area counties and valid US zip codes using the US Department of Housing and Urban Development Zip Code Crosswalk Files [33]. For invalid or missing zip codes, we assigned participants' locations based on latitude and longitude (i.e., GeoIP location, an estimation of the respondent's location based on their IP address), which were mapped to US counties using the US Census Bureau Cartographic Boundary Files [34].

### Level of concern, social distancing behaviors, and difficulties

We considered three outcomes: social distancing behaviors (all of the time, most of the time, some of the time, none of the time); experienced difficulties with daily activities (access to healthcare, childcare, transportation, job loss, or difficulty obtaining essential items including food, medications, and hand sanitizer); and level of concern regarding the COVID-19 crisis

(extremely concerned, very concerned, moderately concerned, somewhat concerned, not at all concerned). These outcome measures were selected for our analysis because we anticipated they would be sensitive enough to capture meaningful changes in behaviors and attitudes in response to COVID-19 at this early point in the natural history of the pandemic, but still represent meaningful impacts on individuals' day-to-day experiences.

## Statistical analysis

We first summarized demographic characteristics for survey respondents living in the seven Bay Area counties affected by the announcement on March 16, 2020 compared to respondents living elsewhere within the U.S.

**Changes before and after the announcement.** We used Yates' continuity-corrected test of proportions to assess changes in levels of social distancing, the proportion of respondents experiencing difficulty with daily activities, and level of concern regarding the COVID-19 crisis after versus before the announcement of the SIPO separately for respondents in the seven Bay Area counties and for respondents elsewhere in the U.S.

**Difference-in-differences estimates.** We used a difference-in-differences (DID) approach with linear probability models to estimate the impact of the SIPO announcement [35,36]. Because the majority of survey responses were collected by March 19, 2020, the DID analysis was focused on examining the impact of the Bay Area SIPO announcement on March 16, 2020. The estimator compared the change in responses after versus before March 16, 2020 among respondents in the Bay Area versus elsewhere in the U.S. The DID approach assumes that any changes that occurred outside of the Bay Area reflect background or secular trends. Under the assumption that these trends would have been parallel among respondents in the Bay Area and elsewhere had the announcement not occurred, the resulting DID estimates correspond to the change in each outcome attributable to the announcement itself in the Bay Area. We calculated DID estimates in the study population overall, and within subgroups defined by gender, age, and household composition (at least one child at home, at least one adult > 65 years).

**Sensitivity analyses.** We conducted the following sensitivity analyses. First, we considered a series of alternative specifications for our main DID analysis: (1) we repeated our main analysis excluding responses after March 19, 2020 at which point California announced a statewide SIPO and when other state SIPOs occurred (**S1 Fig**). (2) We compared responses from the entire state of California to those respondents elsewhere in the U.S. Because the announcement was highly publicized on mainstream news media channels and social media platforms, survey respondents living in California outside of the seven Bay Area counties may have modified their behaviors. (3) Restrictions were announced on March 16, 2020 for Washington state. Therefore, we repeated our main analysis with respondents from Washington state combined with Bay Area respondents.

As an additional sensitivity analysis, we repeated our main analyses estimating marginal probabilities from both logit and probit models as a robustness check, however we prefer linear probability models for our main analysis due to potential issues surrounding non-collapsibility with interaction terms in non-linear models [37].

We conducted all statistical analyses using R version 3.2.3 (R Foundation for Statistical Computing, Vienna, Austria). This study was approved by the Institutional Review Board at Stanford University.

## Results

In total, 22,913 respondents started the survey. We excluded 4,031 respondents who completed less than 80% of the survey, 1,136 respondents with no geolocation data, and 203 international

respondents. The final analytic sample included 17,543 respondents of whom 4,161 (24%) were from the seven Bay Area counties. Among respondents from the Bay Area, 2,951 (70.9%) completed the survey prior to March 16, 2020. Among respondents living elsewhere in the U. S., 8,410 (62.8%) completed the survey prior to March 16, 2020 (**Table 1**), with 90% of survey responses collected by March 19, 2020. (**S1 Fig**) Overall, the majority of respondents were younger than 66 years (N = 90%), and the majority (84%) had earned at least a bachelor's degree. The majority of respondents were female (72%), and most (96%) had some form of health insurance. Approximately 41% of respondents indicated living with at least one child under the age of 18 years and 19% indicated living with at least one adult over the age of 65 years.

Respondents from the Bay Area were less likely to identify as non-Hispanic white as compared with other respondents (73.6% versus 86.0%) and less likely to identify as Black (0.7% versus 1.4%). Respondents from the Bay Area were more likely to be Asian or Pacific Islanders (15.1% versus 4.5%) or Hispanic/Latino (4.9% versus 4.1%). Respondents from the Bay Area were also less likely to be under age 36 years (21.1% versus 31.0%) and slightly more likely to be over age 65 years (13.9% versus 8.6%). The distribution of participants by gender, educational attainment, and household composition was similar among respondents from the Bay Area and respondents living elsewhere. We noted only minor differences between respondents who completed the survey before or after March 16, 2020 in the Bay Area or elsewhere, except for the percentage of respondents who were female and living outside of the Bay Area which was substantially lower before March 16, 2020 versus afterwards (52.5% versus 79.1%). (**S1 Table**)

## Changes before and after the SIPO announcement

In **Table 2**, we present the change in level of social distancing, difficulties experienced, and level of concern following the March 16, 2020 announcement for respondents from the Bay Area and respondents living elsewhere. In general, we observed similar trends in the two groups. We found an increase in the proportion of respondents practicing social distancing all of the time after the announcement in the Bay Area (9.2%, 95% CI: 6.3, 12.1) and elsewhere (3.4%, 95% CI: 2.0, 4.9). We also observed increases in the proportion sheltering in place most of the time among survey respondents from the Bay Area (5.7%, 95% CI: 2.3, 9.0) and elsewhere (8.5%, 95% CI: 6.8, 10.3). The proportion of respondents sheltering in place some of the time and none of the time decreased both among respondents from the Bay Area and elsewhere.

Respondents also reported more difficulty associated with activities such as obtaining food, hand sanitizer, and medications after the March 16, 2020 announcement versus before. The increase in difficulty was largest for obtaining food for both respondents from the Bay Area (13.3%, 95% CI: 10.1, 16.5) and elsewhere (8.2%, 95% CI: 6.6, 9.8). Similarly, both groups reported greater difficulty obtaining hand sanitizer. Greater difficulty with wages was reported more frequently by respondents from the Bay Area following the announcement (4.7%, 95% CI: 2.4, 7.0) and even more so by respondents living elsewhere (6.4%, 95% CI: 5.1, 7.7). Respondents in both groups were also more likely to report difficulty related to job loss following the announcement (Bay Area: 1.2%, 95% CI: 0.3, 2.0; Elsewhere: 1.6%, 95% CI 1.0, 2.1).

We observed only small changes in level of concern regarding the COVID-19 crisis after the March 16, 2020 announcement among respondents in the Bay Area. Among respondents living elsewhere, we observed a decrease in the proportion of respondents reporting they were "extremely concerned" after the announcement (- 4.1%, 95% CI: - 5.7, - 2.4).

**Table 1. Demographic characteristics for Bay Area and in the study population overall–N (%) [1].**

| | Bay Area [2] (N = 4,161) | Elsewhere [2] (N = 13,382) | Overall (N = 17,543) |
|---|---|---|---|
| **Timing of Survey Response** | | | |
| Before 12:00 AM on March 16, 2020 | 2,951 (70.9) | 8,410 (62.8) | 11,361 (64.8) |
| After 12:00 AM on March 16,2020 | 1,210 (29.1) | 4,972 (37.2) | 6,182 (35.2) |
| **Gender** | | | |
| Female | 3,108 (74.7) | 9,450 (70.6) | 12,558 (71.6) |
| Male | 1,015 (24.4) | 3,757 (28.1) | 4,772 (27.2) |
| Other | 27 (0.6) | 142 (1.1) | 169 (1.0) |
| **Race/Ethnicity [3]** | | | |
| Non-Hispanic White | 3,063 (73.6) | 11,503 (86.0) | 14,556 (83.0) |
| Asian and Pacific Islander | 629 (15.1) | 605 (4.5) | 1,234 (7.0) |
| Hispanic/Latino | 204 (4.9) | 544 (4.1) | 748 (4.3) |
| Black | 31 (0.7) | 187 (1.4) | 218 (1.2) |
| Other | 168 (4.0) | 406 (3.0) | 574 (3.3) |
| **Age** | | | |
| < 26 years | 136 (3.3) | 876 (8.6) | 1,012 (5.8) |
| 26–35 years | 741 (17.8) | 2,993 (22.4) | 3,734 (21.3) |
| 36–45 years | 1,029 (24.7) | 3,693 (27.6) | 4,722 (26.9) |
| 46–55 years | 925 (22.2) | 2,736 (20.4) | 3,661 (20.9) |
| 56–65 years | 715 (17.2) | 1,873 (14.0) | 2,588 (14.8) |
| > 65 years | 578 (13.9) | 1,156 (8.6) | 1,734 (9.9) |
| **Education** | | | |
| Less than High School | 8 (0.2) | 39 (0.3) | 47 (0.3) |
| High School or GED | 53 (1.3) | 322 (2.4) | 375 (2.1) |
| Some College | 411 (9.9) | 2,030 (15.2) | 2,441 (13.9) |
| Bachelor's Degree | 3,682 (88.5) | 10,984 (82.1) | 14,666 (83.6) |
| **Health Insurance** | | | |
| Yes | 4,085 (98.2) | 12,832 (95.9) | 16,917 (96.4) |
| No | 58 (1.4) | 490 (3.7) | 548 (3.1) |
| I don't Know | 10 (0.2) | 32 (0.2) | 42 (0.2) |
| **Children in Household (<18 years)** | | | |
| None | 2,296 (55.2) | 7,986 (59.7) | 10,282 (58.6) |
| One | 649 (15.6) | 2,074 (15.5) | 2,723 (15.5) |
| Two | 935 (22.5) | 2,245 (16.8) | 3,180 (18.1) |
| Three or more | 245 (5.9) | 953 (7.1) | 1,198 (6.8) |
| **Senior in Household (>65 years)** | | | |
| None | 3,222 (77.4) | 11,038 (82.3) | 14,260 (81.3) |
| One | 620 (14.9) | 1,528 (11.4) | 2,148 (12.2) |
| Two | 250 (6.0) | 614 (4.6) | 864 (4.9) |
| Three or more | 24 (0.6) | 52 (0.4) | 76 (0.4) |

1. Gender was missing for 44 respondents; race/ethnicity was missing for 203 respondents; age is missing for 92 respondents; educational attainment was missing for 14 respondents; health insurance status was missing for 36 respondents; number of children (< 18 years) in the household was missing for 160 respondents and number of seniors (> 65 years) in household was missing for 195 respondents.

2. Respondents in the Bay Area included those who resided in San Francisco, Santa Clara, San Mateo, Marin, Contra Costa, Alameda, or Santa Cruz county at the time they completed the survey. Respondents elsewhere were those who resided in other California counties or other U.S. states. International respondents were excluded.

3. Asian and Pacific Islander includes respondents who identified as Asian Indian, Chinese, Japanese, Korean, Vietnamese, Filipino, Native Hawaiian, Chamorro, other Pacific Islander, or other Asian.

**Table 2. Changes in social distancing, difficulties, and concern after the shelter-in-place versus before in the Bay Area versus elsewhere in the U.S.**

| | Bay Area | | | Elsewhere | | |
|---|---|---|---|---|---|---|
| | Before–% (N = 2,951) | After–% (N = 1,210) | Percent Change [4] (95% CI) | Before–% (N = 8,410) | After–% (N = 4,972) | Percent Change [4] (95% CI) |
| **Social Distancing [1]** | | | | | | |
| All of the time | 17.3 | 26.5 | 9.21 (6.32, 12.1) | 19.1 | 22.6 | 3.40 (1.95, 4.85) |
| Most of the time | 54.4 | 60.0 | 5.65 (2.29, 9.00) | 48.1 | 56.7 | 8.54 (6.78, 10.3) |
| Some of the time | 26.7 | 12.3 | 14.4 (11.9, 16.9) | 29.4 | 18.6 | - 10.8 (- 12.2, - 9.31) |
| None of the time | 1.6 | 1.2 | - 0.47 (- 1.28, 0.34) | 3.3 | 2.2 | - 1.11 (- 1.69, - 0.54) |
| **Difficulties [2]** | | | | | | |
| Access to Healthcare | 4.2 | 7.4 | 3.19 (1.49, 4.88) | 4.2 | 7.0 | 2.83 (1.98, 3.66) |
| Childcare | 15.1 | 15.3 | 0.18 (- 2.29, 2.64) | 9.4 | 13.1 | 3.61 (2.47, 4.75) |
| Food | 23.8 | 37.1 | 13.3 (10.1, 16.5) | 23.5 | 31.6 | 8.17 (6.57, 9.76) |
| Job Loss | 0.6 | 1.8 | 1.17 (0.31, 2.04) | 1.4 | 2.9 | 1.56 (1.01, 2.10) |
| Medications | 7.6 | 8.9 | 1.34 (- 0.59, 3.26) | 7.1 | 8.6 | 1.52 (0.54, 2.48) |
| Sanitizer | 63.1 | 68.8 | 5.71 (2.52, 8.91) | 59.0 | 62.5 | 3.50 (1.77, 5.22) |
| Transportation | 2.8 | 4.7 | 1.93 (0.54, 3.32) | 3.2 | 3.7 | 0.55 (- 0.11, 1.21) |
| Wages | 9.4 | 14.1 | 4.70 (2.42, 6.98) | 11.3 | 17.7 | 6.41 (5.14, 7.69) |
| **Level of Concern [3]** | | | | | | |
| Extremely concerned | 29.0 | 28.2 | - 0.83 (- 3.90, 2.25) | 32.9 | 28.9 | - 4.05 (- 5.68, - 2.42) |
| Very concerned | 36.9 | 36.1 | - 0.75 (- 4.03, 2.52) | 35.0 | 36.8 | 1.79 (0.09, 3.49) |
| Moderately concerned | 25.6 | 26.3 | 0.73 (- 2.27, 3.73) | 23.1 | 24.8 | 1.75 (0.23, 3.27) |
| A little concerned | 7.5 | 8.2 | 0.73 (- 1.14 2.60) | 7.4 | 8.1 | 0.74 (- 0.22, 1.70) |
| Not at all concerned | 1.1 | 1.2 | 0.12 (- 0.67, 0.91) | 1.6 | 1.4 | - 0.24 (- 0.67, 0.20) |

1. Respondents were asked to select their level of social distancing. We created a mutually exclusive set of indicator variables.

2. Respondents were asked to select all of the difficulties they had experienced because of the COVID-19 crisis; categories are not mutually exclusive.

3. Respondents were asked to select their level of concern regarding the COVID-19 crisis. We created a mutually exclusive set of indicator variables.

4. We calculated the change in level of concern, social distancing levels, and experienced difficulties before and after the shelter-in-place announcement with 95% confidence intervals using Yates' corrected test of proportions.

### Difference-in-differences estimates

In **Table 3,** we present DID estimates for the change in the proportion of respondents who were social distancing all of the time after the announcement in the Bay Area versus elsewhere. Overall, the proportion of respondents social distancing all of the time increased after the announcement in the Bay Area versus elsewhere (5.8%, 95% CI: 2.8, 8.8). Relative increases were greatest among men (9.3%, 95% CI: 3.2, 15.4), adults between the ages of 46 and 65 years (6.7%, 95% CI 1.8, 11.7), and respondents from households with children. We calculated DID estimates for experienced difficulties in the Bay Area versus elsewhere following the announcement. We noted the strongest differences for difficulty obtaining food (5.2%, 95% CI: 1.8, 8.5), followed by difficulty with transportation (2.2, 95% CI: - 1.5, 5.9) (**Fig 1, S2 Table**). We observed limited evidence of increased difficulty with healthcare, obtaining hand sanitizer, or obtaining medications.

In **Table 4** we present DID estimates for the change in the proportion of respondents who were extremely concerned following the announcement in the Bay Area versus elsewhere. Overall, the proportion of respondents who reported extreme worry did not increase after the announcement for most groups, with the exception of those aged 46–65 years (8.03, 95% CI 2.03, 14.0) and respondents living with at least one child (6.20, 95% CI 0.62, 11.8). The

**Table 3. Percentage of respondents who were social distancing all of the time in Bay Area versus elsewhere in the U.S. before and after the March 16th, 2020 Bay Area Shelter-in-Place Announcement and difference-in-differences estimates for the study population overall and within strata of gender, age category, and household composition[1].**

| | Bay Area | | Elsewhere | | DID Estimate (95% CI) |
|---|---|---|---|---|---|
| | Before–% (N = 2,951) | After–% (N = 1,210) | Before–% (N = 8,410) | After–% (N = 4,972) | |
| **Overall [2]** | 511 (17.3) | 321 (26.5) | 1,610 (19.1) | 1,121 (22.5) | 5.81 (2.78, 8.84) |
| **Sex [3]** | | | | | |
| Women | 392 (18.0) | 244 (26.3) | 1,083 (19.6) | 915 (23.3) | 4.62 (1.08, 8.16) |
| Men | 113 (15.1) | 70 (26.1) | 501 (18.1) | 194 (19.7) | 9.32 (3.22, 15.4) |
| **Age Category [4]** | | | | | |
| 25 Years or Less | 14 (16.7) | 10 (19.2) | 69 (11.5) | 42 (15.2) | - 1.12 (- 13.9, 11.6) |
| 26–45 Years | 217 (18.0) | 149 (26.3) | 782 (19.0) | 577 (22.5) | 4.72 (0.24, 9.20) |
| 46–65 Years | 181 (14.9) | 96 (22.4) | 608 (20.1) | 330 (20.9) | 6.72 (1.75, 11.7) |
| Older than 65 Years | 91 (21.5) | 65 (41.9) | 149 (23.7) | 168 (31.9) | 12.1 (2.53, 21.7) |
| **Household Composition [5]** | | | | | |
| Households with child < 18 | 240 (17.5) | 132 (29.1) | 679 (20.5) | 448 (23.2) | 8.99 (4.09, 13.9) |
| Households with adult > 65 | 115 (18.3) | 80 (30.2) | 306 (21.9) | 222 (27.9) | 5.91 (- 1.18, 13.0) |

**1.** We used a difference-in-difference in estimator that compared the change in response following the March 16, 2020 shelter-in-place announcement in The San Francisco Bay Area versus elsewhere. We calculated the percent change in California vs. elsewhere by multiplying linear probability estimates by 100.

**2.** Percentages and DID estimate for the study population overall (N = 17,543).

**3.** Percentages and DID estimates for subgroup of women (N = 12,558) and men (N = 4,772).

**4.** Percentages and DID estimates among respondents less than 25 years old (N = 744), between the ages 25 and 34 (N = 3,493), between the ages of 35 and 44 (N = 4,807), between the ages of 45 and 54 (N = 3,790), between the ages of 55 and 64 (N = 2,679) and 65 years or older (N = 1,938).

**5.** Percentages and DID estimates for household with at least one child (N = 7,261) and at least one elderly household member (N = 3,283).

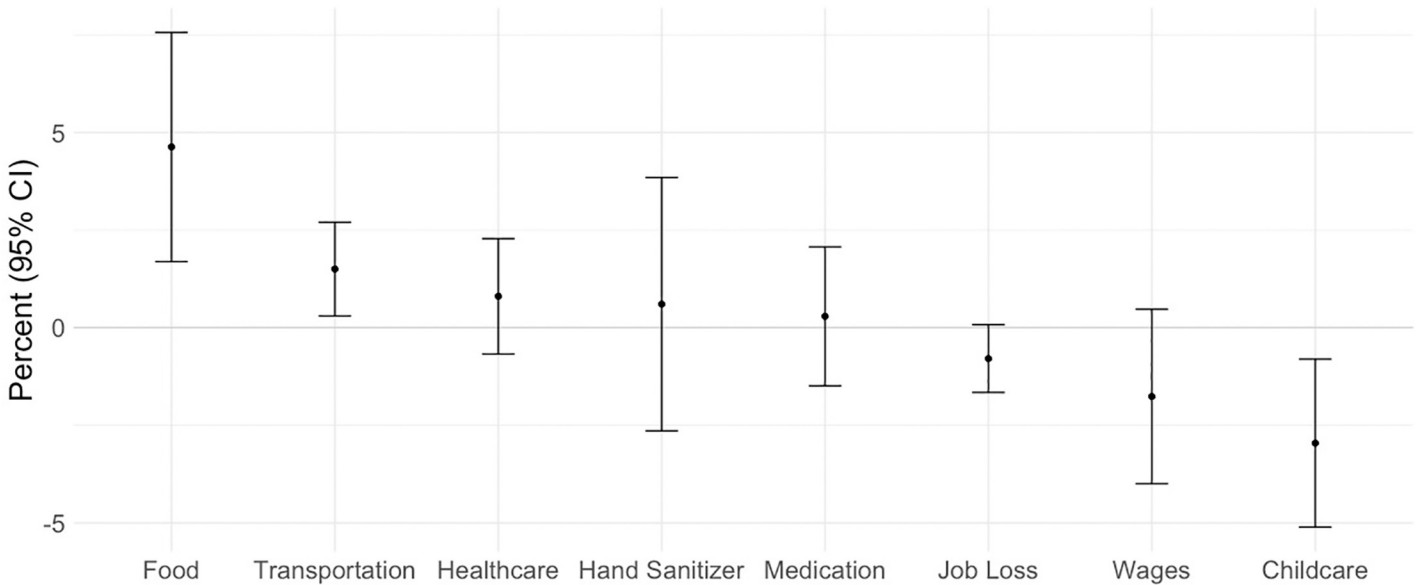

**Fig 1. Difference-in-difference estimates for experienced difficulties in the Bay Area versus elsewhere following the March 16, 2020 announcement of the Bay Area shelter in place order.** We used linear probability models to estimate the change in the San Francisco Bay Area versus elsewhere for each of the above experienced difficulties for the full sample (N = 17,543) and among the subset of a respondents living in a household with a child < 18 for difficulty with childcare (N = 7,062). We transformed model coefficients into percentages by multiplying estimated proportions by 100%.

**Table 4. Percentage of respondents who were extremely worried about the COVID-19 crisis in the Bay Area and elsewhere in the U.S. before and after the March 16[th], 2020 Bay Area shelter-in-place announcement and difference-in-differences estimates for the study population overall and within strata of gender, age category, and household composition[1].**

| | Bay Area | | Elsewhere | | DID Estimate (95% CI) |
|---|---|---|---|---|---|
| | Before–% (N = 2,951) | After–% (N = 1,210) | Before–% (N = 8,410) | After–% (N = 4,972) | |
| **Overall [2]** | 856 (29.0) | 341 (28.2) | 2,768 (32.9) | 1,435 (28.9) | 3.23 (- 0.26, 6.71) |
| **Sex [3]** | | | | | |
| Women | 643 (29.5) | 280 (30.1) | 1,880 (34.1) | 1,204 (30.6) | 4.07 (0.01, 8.12) |
| Men | 205 (27.4) | 55 (20.5) | 851 (30.7) | 218 (22.2) | 1.58 (- 5.45, 8.61) |
| **Age Category [4]** | | | | | |
| 25 Years or Less | 13 (15.5) | 6 (11.5) | 124 (20.7) | 39 (14.1) | 2.60 (- 1.17, 16.9) |
| 26–45 Years | 267 (22.2) | 146 (25.7) | 1,281 (31.1) | 673 (26.2) | 0.09 (- 4.97, 5.15) |
| 46–65 Years | 355 (29.3) | 148 (34.6) | 1,143 (37.8) | 554 (35.0) | 8.03 (2.03, 14.0) |
| Older than 65 Years | 113 (26.7) | 40 (25.8) | 210 (33.3) | 161 (30.6) | 1.82 (- 8.16, 11.8) |
| **Household Composition [5]** | | | | | |
| Households with child < 18 | 421 (30.6) | 146 (32.2) | 1,105 (33.4) | 556 (28.8) | 6.20 (0.62, 11.8) |
| Households with adult > 65 | 179 (28.5) | 74 (27.9) | 511 (33.6) | 268 (33.7) | 2.35 (- 5.55, 10.3) |

1. We used a difference-in-difference in estimator that compared the change in response following the March 16, 2020 shelter-in-place announcement in the San Francisco Bay Area versus elsewhere. We calculated the percent change in California vs. elsewhere by multiplying linear probability estimates by 100.

2. Percentages and DID estimates for the study population overall (N = 17,543).

3. Percentages and DID estimates for subgroup of women (N = 12,558) and men (N = 4,772).

4. Percentages and DID estimates among respondents less than 25 years old (N = 744), between the ages 25 and 34 (N = 3,493), between the ages of 35 and 44 (N = 4,807), between the ages of 45 and 54 (N = 3,790), between the ages of 55 and 64 (N = 2,679) and 65 years or older (N = 1,938).

5. Percentages and DID estimates for household with at least one child (N = 7,261) and at least one elderly household member (N = 3,283).

proportion reporting extreme worry decreased in some groups including men, those under age 25, and those living outside the Bay Area.

## Sensitivity analyses

Across alternative specifications of our main analysis, the overall pattern remained consistent. Findings were slightly accentuated when we excluded survey responses after March 19, 2020, and slightly attenuated when we compared California respondents to respondents elsewhere in the U.S. or when we combined Washington state respondents with Bay Area Respondents. (**Tables 5–7**) Results from robustness checks estimating marginal probabilities using logit and probit models are presented in **Tables 8–10**. The overall pattern of our robustness checks are consistent with those of our primary analysis.

## Discussion

We examined changes in attitudes and behaviors in response to the announcement of the nation's first SIPO within a cross-sectional convenience sample of 17,543 respondents living in the San Francisco Bay Area and elsewhere in the U.S. recruited through three social media platforms. Differences in key demographic characteristics (level of insurance, educational attainment, race/ethnicity) preclude generalization of our findings to the Bay Area or to the U.S. more broadly. Nevertheless, the present study contributes meaningfully to the growing literature on the impacts of SIPOs by capturing not only how levels of sheltering-in-place changed following the announcement, but also by characterizing how difficulties with daily activities and levels of concern regarding how COVID-19 may have changed in the days that

**Table 5. Alternative characterizations of DID groups for analysis of respondents who were extremely worried about COVID-19 after versus before the San Francisco Bay Area shelter-in-place announcement.**

| | Alternative 1 [1] ß (95% CI) [4] | Alternative 2 [2] ß (95% CI) [4] | Alternative 3 [3] ß (95% CI) [4] |
|---|---|---|---|
| **Overall [5]** | 1.14 (- 2.79, 5.07) | 2.67 (- 0.40, 5.75) | 2.80 (- 0.42, 6.03) |
| **Sex [6]** | | | |
| Women | 2.41 (- 2.14, 6.95) | 5.17 (4.81, 5.53) | 3.22 (- 0.52, 6.96) |
| Men | - 1.58 (- 9.64, 6.48) | - 2.98 (- 9.11, 3.15) | 1.77 (- 4.85, 8.38) |
| **Age Category [7]** | | | |
| < 25 Years | 3.45 (- 12.1, 19.0) | - 0.90 (- 12.9, 11.0) | 0.32 (- 13.0, 13.6) |
| 26–45 Years | 1.22 (- 4.49, 6.93) | - 0.23 (- 4.69, 4.22) | 0.06 (- 4.58, 4.71) |
| 46–65 Years | 1.38 (- 5.52, 8.28) | 4.58 (- 0.74, 9.91) | 7.10 (1.55, 12.6) |
| > 65 Years | - 1.28 (- 12.0, 9.47) | 7.74 (- 1.32, 16.8) | 1.93 (- 7.48, 11.3) |
| **Household Composition [8]** | | | |
| Households with Children < 18 | 5.08 (- 1.21, 11.4) | 5.03 (0.03, 10.0) | 5.32 (0.16, 10.5) |
| Households with Senior > 65 | 1.25 (- 7.60, 10.1) | 5.61 (- 1.55, 12.8) | 0.83 –(6.65, 8.31) |

1. For alternative 1, we used a DID estimator that compared Bay Area versus elsewhere with follow-up restricted to responses prior to March 20, 2020.

2. For alternative 2, we used a DID estimator that compared respondents in California versus respondents elsewhere.

3. For alternative 3, we used a DID estimator that compared respondents in the Bay Area **and** Washington state versus respondents elsewhere.

4. We used linear probability models and calculated the percent change in California vs. elsewhere by multiplying linear probability estimates by 100.

5. DID estimate for the study population overall (N = 17,543).

6. DID estimates for subgroup of women (N = 12,558) and men (N = 4,772).

7. DID estimates among respondents less than 25 years old (N = 744), between the ages 25 and 34 (N = 3,493), between the ages of 35 and 44 (N = 4,807), between the ages of 45 and 54 (N = 3,790), between the ages of 55 and 64 (N = 2,679) and 65 years or older (N = 1,938).

8. DID estimates among household with at least one child (N = 7,261) and at least one elderly household member (N = 3,283).

immediately preceded and immediately followed the announcement of the nation's first SIPO for seven Bay Area counties.

This announcement occurred at a point where the seriousness of the COVID-19 pandemic for the U.S. was increasingly recognized, but the eventual national impact was yet to be realized [38–41]. As such, the results of this study offer some insight into our collective disposition towards the pandemic at a unique point in history as the very first decisions to implement SIPOs were made. As local and state governments, school districts and universities, and other governing bodies begin to enact, rollback, and re-enact similar SIPOs and nonpharmaceutical interventions, our findings may help quantify the impact of these orders to better inform decisionmakers.

Much of the literature-to-date on orders to shelter-in-place from the U.S. examines their effectiveness, and generally demonstrates short-term behavior change [32,42,43]. Difference-in-differences analysis of daily state-level data from March 8, 2020–April 17, 2020 demonstrates that enactment of SIPOs is associate with an approximate 2.1 percent increase in the stay-at-home rate nationally [29]. Using data form the U.S. Department of Transportation, Gupta and colleagues find that out-of-state travel fell by approximately 54% between March 1, 2020–April 14, 2020 [43]. Consistent with these findings, analyses of anonymized mobile phone data suggest substantial reductions in mobility [44,45].

Apparent consequences of compliance with SIPOs include psychiatric distress and social isolation. For example, data collected from 986 San Francisco Bay Area residents participating in the ongoing Stanford Well for LIFE Study demonstrated an eight-fold increase in the proportion of participants who reported feeling distressed [46]. Similarly, in a nationwide online

**Table 6. Alternative characterizations of DID groups for analysis of respondents who were sheltering-in-place all of the time after versus before the San Francisco Bay Area shelter-in-place announcement.**

| | Alternative 1 [1] ß (95% CI) [4] | Alternative 2 [2] ß (95% CI) [4] | Alternative 3 [3] ß (95% CI) [4] |
|---|---|---|---|
| **Overall [5]** | 5.97 (2.58, 9.37) | 3.29 (0.49, 6.10) | 3.71 (1.03, 6.38) |
| **Sex [6]** | | | |
| Women | 4.96 (1.02, 8.90) | 2.41 (- 0.85, 5.67) | 3.32 (0.18, 6.47) |
| Men | 8.82 (1.88, 15.8) | 6.57 (0.84, 12.3) | 5.41 (0.10, 10.7) |
| **Age Category [7]** | | | |
| < 25 Years | 1.57 (- 12.1, 15.3) | 1.01 (- 10.8, 12.8) | - 1.93 (- 12.5, 8.65) |
| 26–45 Years | 5.67 (0.66, 10.7) | 0.69 (- 3.42, 4.81) | 2.18 (- 1.77, 6.13) |
| 46–65 Years | 3.34 (- 2.36, 9.04) | 5.80 (1.20, 10.4) | 4.27 (- 0.15, 8.68) |
| > 65 Years | 15.9 (5.61, 26.2) | 9.12 (0.06, 18.2) | 11.4 (2.67, 20.1) |
| **Household Composition [8]** | | | |
| Households with Children < 18 | 8.80 (3.32, 14.3) | 5.93 (3.52, 8.35) | 5.70 (1.31, 10.1) |
| Households with Senior > 65 | 7.42 (- 0.44, 15.3) | 4.97 (- 1.74, 11.7) | 1.55 (- 4.87, 7.97) |

**1.** For alternative 1, we used a DID estimator that compared Bay Area versus elsewhere with follow-up restricted to responses prior to March 20, 2020.

**2.** For alternative 2, we used a DID estimator that compared respondents in California versus respondents elsewhere.

**3.** For alternative 3, we used a DID estimator that compared respondents in the Bay Area **and** Washington state versus respondents elsewhere.

**4.** We used linear probability models and calculated the percent change in California vs. elsewhere by multiplying linear probability estimates by 100.

**5.** DID estimate for the study population overall (N = 17,543).

**6.** DID estimates for subgroup of women (N = 12,558) and men (N = 4,772).

**7.** DID estimates among respondents less than 25 years old (N = 744), between the ages 25 and 34 (N = 3,493), between the ages of 35 and 44 (N = 4,807), between the ages of 45 and 54 (N = 3,790), between the ages of 55 and 64 (N = 2,679) and 65 years or older (N = 1,938).

**8.** DID estimates among household with at least one child (N = 7,261) and at least one elderly household member (N = 3,283).

sample of 435 U.S. adults conducted in March of 2020 respondents reported symptoms of depression, generalized anxiety, stress, and insomnia in associating with stay-at-home orders [47]. Studies-to-date also suggest increasing suicidal ideation among those under lockdown or sheltering in place [48], although evidence remains mixed [49].

In the present study, we found that participants' behaviors and attitudes regarding the COVID-19 pandemic evolved even within our brief survey period. After the SIPO was

**Table 7. Alternative characterization of DID groups for analysis of experienced difficulties after versus before the San Francisco Bay Area shelter-in-place announcement.**

| | Alternative 1 [1] ß (95% CI) [4] | Alternative 2 [2] ß (95% CI) [4] | Alternative 3 [3] ß (95% CI) [4] |
|---|---|---|---|
| **Food** | 6.62 (2.88, 10.4) | 4.63 (1.69, 7.56) | 3.23 (0.16, 6.31) |
| **Transportation** | 1.05 (- 0.47, 2.67) | 1.50 (0.30, 2.70) | 0.55 (- 0.70, 1.81) |
| **Healthcare** | 0.85 (- 0.96, 2.67) | 0.80 (- 0.67, 2.28) | 0.27 (- 1.28, 1.81) |
| **Hand Sanitizer** | 3.29 (- 0.89, 7.48) | 0.60 (- 2.64, 3.85) | 1.02 (- 2.37, 4.42) |
| **Medication** | 0.05 (- 2.22, 2.33) | 2.90 (- 1.49, 2.09) | - 0.18 (- 2.04, 1.68) |
| **Job Loss** | - 0.56 (- 1.61, 0.50) | - 0.79 (- 1.66, 0.08) | - 0.61 (- 1.52, 0.30) |
| **Childcare** | - 4.66 (- 10.7, 1.38) | - 0.99 (- 5.81, 3.84) | - 4.41 (- 9.38, 0.57) |
| **Wages** | - 1.41 (- 4.23, 1.41) | - 1.76 (- 4.00, 0.47) | - 2.33 (- 4.66, 0.01 |

**1.** For alternative 1, we used a DID estimator that compared Bay Area versus elsewhere with follow-up restricted to responses prior to March 20, 2020.

**2.** For alternative 2, we used a DID estimator that compared respondents in California versus respondents elsewhere.

**3.** For alternative 3, we used a DID estimator that compared respondents in the Bay Area **and** Washington state versus respondents elsewhere.

**4.** We used linear probability models and calculated the percent change in California vs. elsewhere by multiplying linear probability estimates by 100.

**Table 8. Robustness check with marginal probabilities estimated from logit and probit models for respondents who were extremely worried about COVID-19 after versus before the San Francisco Bay Area shelter-in-place announcement [1].**

|  | Logit Estimates [2] 95% CI | Probit Estimates [3] 95% CI |
|---|---|---|
| **Overall [4]** | 0.033 (- 0.004, 0.070) | 0.033 (- 0.004, 0.069) |
| **Sex [5]** |  |  |
| Women | 0.042 (- 0.001, 0.085) | 0.042 (-0.001, 0.084) |
| Men | 0.012 (- 0.066, 0.089) | 0.013 (-0.062, 0.088) |
| **Age Category [6]** |  |  |
| < 25 Years | 0.018 (-0.154, 0.191) | 0.020 (-0.149, 0.184) |
| 26–45 Years | 0.0003 (-0.051, 0.052) | 0.0005 (-0.051, 0.052) |
| 46–65 Years | 0.086 (0.021, 0.151) | 0.085 (0.021, 0.148) |
| > 65 Years | 0.017 (- 0.088, 0.122) | 0.017 (-0.085, 0.121) |
| **Household Composition [7]** |  |  |
| Households with Children < 18 | 0.065 (0.005, 0.125) | 0.064 (0.005, 0.124) |
| Households with Senior > 65 | 0.023 (- 0.061, 0.107) | 0.023 (-0.060, 0.105) |

1. We used a difference-in-difference in estimator that compared the change in response following the March 16, 2020 shelter-in-place announcement in the San Francisco Bay Area versus elsewhere.

2. We estimated the marginal probabilities using logit models.

3. We estimated the marginal probabilities using probit models.

4. DID estimate for the study population overall (N = 17,543).

5. DID estimates for subgroup of women (N = 12,558) and men (N = 4,772).

6. DID estimates among respondents less than 25 years old (N = 744), between the ages 25 and 34 (N = 3,493), between the ages of 35 and 44 (N = 4,807), between the ages of 45 and 54 (N = 3,790), between the ages of 55 and 64 (N = 2,679) and 65 years or older (N = 1,938).

7. DID estimates among household with at least one child (N = 7,261) and at least one elderly household member (N = 3,283).

announced for the Bay Area, social distancing increased. Increases in level of social distancing were more pronounced among respondents in the Bay Area versus those living elsewhere in the U.S., adults older than 46 years, and those living with children or an adult over age 65 years. This pattern may be explained by early suspicions that older adults were most vulnerable to COVID-19 [50].

Respondents were most likely to report difficulty obtaining food, with increases in difficulty obtaining food more pronounced in the Bay Area following the announcement of the SIPO. Difficulties obtaining food are most likely due to increased demand from consumers (rather than supply-side issues) as suggested by various media reports [51–53]. Increases in difficulty with access to healthcare, hand sanitizer, and transportation were similar among respondents in the Bay Area versus those living elsewhere. We detected the early impacts on job loss and wages, which were followed by a national surge in unemployment after the study period [54,55]. We anticipate that our findings may further underestimate the impacts of SIPOs on job loss and wages given the high levels of educational attainment in our study population, as may respondents may have been able to transition more easily to remote work [56].

Finally, we found that approximately one-third of respondents were "extremely concerned" about the COVID-19 crisis, although we found little evidence to support the idea that levels of concern increased–among respondents in the Bay Area or elsewhere–following the announcement of the SIPO. This raises the interesting question as to whether announcements regarding COVID-19 lead to increased or decreased levels of concern and anxiety that should be considered further in more representative study populations and as the pandemic continues to evolve.

**Table 9. Robustness check with marginal probabilities estimated from logit and probit models for respondents who were sheltering-in-place all of the time after versus before the San Francisco Bay Area shelter-in-place announcement [1].**

| | Logit Estimates [2] 95% CI | Probit Estimates [3] 95% CI |
|---|---|---|
| **Overall [4]** | 0.059 (0.025, 0.093) | 0.059 (0.025, 0.093) |
| **Sex [5]** | | |
| Women | 0.047 (0.008, 0.086) | 0.047 (0.009, 0.085) |
| Men | 0.100 (0.024 0.176) | 0.099 (0.025, 0.171) |
| **Age Category [6]** | | |
| < 25 Years | - 0.016 (- 0.118, 0.086) | - 0.015 (- 0.125, 0.094) |
| 26–45 Years | 0.046 (- 0.003, 0.95) | 0.046 (- 0.002, 0.95) |
| 46–65 Years | 0.079 (0.018, 0.139) | 0.076 (0.018, 0.134) |
| > 65 Years | 0.119 (0.010, 0.228) | 0.0120 (0.012, 0.227) |
| **Household Composition [7]** | | |
| Households with Children < 18 | 0.095 (0.037, 0.153) | 0.094 (0.037, 0.150) |
| Households with Senior > 65 | 0.064 (- 0. 014, 0.143) | 0.063 (-0.014, 0.141) |

**1.** We used a difference-in-difference in estimator that compared the change in response following the March 16, 2020 shelter-in-place announcement in the San Francisco Bay Area versus elsewhere.

**2.** We estimated the marginal probabilities using logit models.

**3.** We estimated the marginal probabilities using probit models.

**4.** DID estimate for the study population overall (N = 17,543).

**5.** DID estimates for subgroup of women (N = 12,558) and men (N = 4,772).

**6.** DID estimates among respondents less than 25 years old (N = 744), between the ages 25 and 34 (N = 3,493), between the ages of 35 and 44 (N = 4,807), between the ages of 45 and 54 (N = 3,790), between the ages of 55 and 64 (N = 2,679) and 65 years or older (N = 1,938).

**7.** DID estimates among household with at least one child (N = 7,261) and at least one elderly household member (N = 3,283).

## Limitations

Despite the large number of survey respondents, older adults, Black respondents, and men were underrepresented in this convenience sample. Similarly, household structure of respondents suggests that a large number of respondents did not have children or elderly family

**Table 10. Robustness check with marginal probabilities estimated from logit and probit models for experienced difficulties after versus before the San Francisco Bay Area shelter-in-place announcement t[1].**

| | Logit Estimates [2] 95% CI | Probit Estimates [3] 95% CI |
|---|---|---|
| **Food** | 0.048 (0.011, 0.081) | 0.047 (0.012, 0.082) |
| **Transportation** | 0.014 (- 0.003, 0.032) | 0.014 (- 0.002, 0.031) |
| **Healthcare** | 0.003 (- 0.013, 0.019) | 0.003 (- 0.013, 0.019) |
| **Hand Sanitizer** | 0.025 (- 0.012, 0.062) | 0.025 (- 0.012, 0.061) |
| **Medication** | - 0.002 (- 0.021, 0.016) | - 0.002 (- 0.022, 0.017) |
| **Job Loss** | 0.005 (- 0.008, 0.017) | 0.003 (- 0.008, 0.015) |
| **Childcare** | - 0.030 (- 0.078, 0.018) | - 0.028 (- 0.078, 0.022) |
| **Wages** | - 0.007 (- 0.031, 0.017) | - 0.009 (- 0.033, 0.015) |

**1.** We used a difference-in-difference in estimator that compared the change in response following the March 16, 2020 shelter-in-place announcement in the San Francisco Bay Area versus elsewhere.

**2.** We estimated the marginal probabilities using logit models.

**3.** We estimated the marginal probabilities using probit models.

members that may have required extra care. Recruitment was convenience sampling via three social media websites. Snowball sampling (through re-posts on Facebook and Twitter) may have further propagated participation among a more homogenous group of respondents. Our results therefore likely underrepresent the true extent of challenges associated with the pandemic across the U.S. and precludes meaningful examination of the early impacts of SIPOs on economically marginalized and vulnerable population subgroups [57,58].

The cross-sectional nature of this study represents an additional limitation. Because we did not observe changes in social distancing, experienced difficulties, and levels of concern in individuals over time, it is possible that our findings are explained at least in part by compositional effects (i.e., systematic differences in respondents who completed the survey before and after March 16th). Reassuringly, we found limited evidence of systematic differences in measured characteristics before and after the March 16th cutoff with the exception of the gender breakdown among respondents who resided outside of the Bay Area.

Although the Bay Area was the first to announce a SIPO nationally, other states and localities introduced SIPOs throughout late March and early April of 2020. However, the highly imbalanced nature of our study sample (over 90% of survey responses were collected by March 19, 2020) precludes meaningful examination of the phased implementation of SIPOs using a staggered difference-in-difference approach. We anticipate that such an approach would yield less precise estimates while necessitating stronger assumptions (e.g., that there was no growing concern as the number of state orders). While the analytic approach presented in our study provides information only regarding the impact of the Bay Area, we ultimately feel it is the most appropriate approach.

Finally, the announcement of SIPO for the seven Bay Area counties was covered extensively in the national media, which makes spillover effects of the announcement to survey respondents living outside of the Bay Area–particularly elsewhere in California–likely. The assumptions of DID are therefore unlikely to be met, and our estimates are more appropriately interpreted as summary measures of the change in the Bay Area relative to the change elsewhere in the U.S. rather than causal estimates of the impact of the announcement. However, in sensitivity analyses to examine spillover in Washington and California and in our sensitivity analysis that excludes survey responses after March 19, 2020 (when additional SIPOs were implemented nationally), we found similar pattern of findings across subgroups of interest.

## Conclusions

We found evidence of increased social distancing and difficulty with daily activities such as food and transportation in the wake of the announcement of the nation's first SIPO, particularly among respondents in the Bay Area. Levels of concern remained fairly consistent throughout the study period among respondents in the Bay Area and elsewhere. Given that our study population was highly educated, concentrated in one of the more affluent areas in the U.S., and queried relatively early in the COVID-19 pandemic, we anticipate that our findings underestimate substantially the impact of county- and statewide SIPOs. As such, our study represents a first step towards understanding the social attitudes and consequences of this crisis. Further research that specifically examines social, economic, and health impacts of COVID-19 especially among vulnerable populations is needed.

## Supporting information

**S1 Fig. Timing of survey responses and statewide shelter-in-place orders.** We depict the frequency of survey responses in the Bay Area and elsewhere in the U.S. by date (top panel); cumulative survey responses by date as percentages (middle panel); and the timing of statewide

shelter-in-place orders (bottom panel).
(DOCX)

**S1 Table. Demographics for the Bay Area and other U.S. states before and after the March 16, 2020 shelter-in-place announcement–N (%) [1].**
(DOCX)

**S2 Table. DID estimates for experienced difficulties in California versus elsewhere following the March 16, 2020 announcement of the Bay Area shelter in place order [1].**
(DOCX)

## Author Contributions

**Conceptualization:** Eleni Linos.

**Formal analysis:** Holly Elser, Mathew V. Kiang.

**Methodology:** Eleni Linos.

**Resources:** Eleni Linos.

**Supervision:** Eleni Linos.

**Writing – original draft:** Holly Elser, Mathew V. Kiang.

**Writing – review & editing:** Holly Elser, Mathew V. Kiang, Esther M. John, Julia F. Simard, Melissa Bondy, Lorene M. Nelson, Wei-ting Chen, Eleni Linos.

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
