## [Decision Letter · Decision Letter 0]

10 Aug 2020

PONE-D-20-20290

Implications of the COVID-19 San Francisco Bay Area Shelter-in-Place Announcement: A Cross-Sectional Social Media Survey

PLOS ONE

Dear Dr. Linos,

Thank you for submitting your manuscript to PLOS ONE. After careful consideration, we feel that it has merit but does not fully meet PLOS ONE’s publication criteria as it currently stands. Therefore, we invite you to submit a revised version of the manuscript that addresses the points raised during the review process.

Both referees found the paper interesting, well-written and relevant. However, R2 has requested a number of revisions while R1 is less enthusiastic. R1 is particularly concerned about the construction of the study sample and composition of the "control group". I also share this concern. Nonetheless, I'd like to give you the opportunity to respond, revise, and fully engage with referee comments. 

We look forward to receiving your revised manuscript.

Kind regards,

M Niaz Asadullah

Academic Editor

PLOS ONE

Journal Requirements:

Reviewers' comments:

Reviewer's Responses to Questions

**Comments to the Author**

1. Is the manuscript technically sound, and do the data support the conclusions?

Reviewer #1: Partly

Reviewer #2: Partly

2. Has the statistical analysis been performed appropriately and rigorously? 

Reviewer #1: No

Reviewer #2: No

3. Have the authors made all data underlying the findings in their manuscript fully available?

Reviewer #1: Yes

Reviewer #2: Yes

4. Is the manuscript presented in an intelligible fashion and written in standard English?

Reviewer #1: Yes

Reviewer #2: Yes

5. Review Comments to the Author

Reviewer #1: The authors present difference-in-difference estimates of the effect of the shelter-in-place (herein "lockdown") orders issued by seven U.S. counties around and including San Francisco, California, on survey respondents’ concerns and self-reported social distancing behavior. In their DID analysis, they find that Bay Area respondents are more likely to report social distancing “all of the time” (save for the youngest respondents, who actually social distance less than non-Bay-Area youth), and that Bay Area respondents post-lockdown had a harder time finding food relative, but otherwise had statistically insignificantly different challenges than non-Bay-Area respondents post-lockdown. They emphasize a number of limitations at the end of their study, all of which point towards their findings being an underestimate of the effect of lockdowns on individuals’ attitudes and behaviors.

I find this paper interesting, but not as well executed as it should be. My two main concerns are the sampling technique (and the subsequent sample it provided, which the authors point out without dissembling) and the definition of their difference-in-difference.

First, the sampling methodology: the authors use snowball sampling on three social media platforms over a period of just over two weeks. They end up with many thousands of responses who are deeply unrepresentative of the United States. To their credit, they neither paper over this fact nor neglect to mention it, but make it plain for the reader. And I agree with them that their sample (especially the educational background of their respondents) means that they likely underestimate the impact of the lockdown on the challenges people faced from sheltering in place. That is precisely why the authors should have chosen a different sampling approach from the get-go.

That said, there’s nothing the authors can do about this now, and they have acknowledged it as much as they can.

My second concern is something the authors can do something about. The authors analyze their data as if there is one and only one lockdown in all of the U.S.: the seven Bay Area counties start sheltering in place on March 16th. To put it bluntly: WHAT? Just three days later, *all of California* goes into lockdown. One week later, 40 percent of the U.S. by population is on lockdown, and just eleven days after the Bay Area begins sheltering in place, half of the United States shelters in place alongside it.

[See https://www.usatoday.com/story/news/nation/2020/03/30/coronavirus-stay-home-shelter-in-place-orders-by-state/5092413002/ for the list of lockdowns—I’m using the 2019 Census estimates of state population, and only including whole state lockdowns, which leaves out, e.g., both the major cities and nearly all the population of Pennsylvania.]

It seems to me that this is a pretty substantial oversight, thought as with the sampling approach, is one that leads the authors to understate potential differences between locked-down and non-locked-down individuals (as it treats many of the lockdown-treated as untreated). It also seems to me that the authors should instead be using something like a staggered difference-in-difference approach, given how many of their respondents likely (a) come from California and (b) come from other locked down states (because of the snowball sampling approach).

In short: it is easy, four months later, to complain about an imperfect approach to getting respondents in the middle of a global pandemic. But inasmuch as PLOS One is less about getting exciting results and more about using good approaches (whatever results these may yield), both the sampling technique and the analysis do not meet that benchmark. The former is not fixable, but the latter is, and fixing it, I think, would improve the paper.

Reviewer #2: Comments on “Implications of the COVID-19 San Francisco Bay Area Shelter-in-Place Announcement: A Cross-Sectional Social Media Survey”

Synopsis

----------

The paper, in my view, aims to share some very timely outcomes and analyses on impacts of COVID-19 in California, which was one of the first states to impose strict social distancing measure. The authors rightly zeroed on the seven counties who imposed the “shelter-in-place” policies. The authors looked at a number of outcomes using a quasi-experimental method (difference-in-difference, p.11). The DID estimates are somewhat modest with the largest impacts on increased difficulties with procuring food and mobility (transportation, p.14, also Table A2 and Figure 1). This may have important implications for lockdown policies in the US and elsewhere.

Comments

------------

1. As a reader, I struggled to understand what the focus of the paper is, in terms of outcomes. The implications and focus of the title, the abstract and the introduction are not specific. The authors should consider what outcomes they are interested in and why.

2. The description of the method is confusing. It is definitely a cross-sectional in the sense the data was collected at the individual levels at one point in time. However, it seems the authors managed to synthesize a panel by aggregating responses before and after imposition of the shelter-in-place policies by counties or geographic locations. This process is not very well described.

3. Related to #2 above, it is important to let the reader know timing of different events. When did the government announce the shelter-in-place policy? How much in advance did the researchers and the citizens know about the policy? This is important because people may have time to adjust their behaviors and that can mitigate some of the outcomes that we eventually see in the study, for example, small size impacts on the outcomes the authors are interested in (again see Figure 1).

4. On page 14 (the manuscript does not have page numbers!), authors report difficulty in obtaining food. However, we don’t get a sense of the mechanism here. As people felt difficulty in moving from one place to another (another outcome the authors report), it is possible there was disruption in supply chain. Some discussion on this would be useful.

5. On page 11, LPM is all good. But I struggled understanding whether the same model can be applicable to all outcomes. Typically, they are applicable for binary outcomes and as robustness checks other models are used such as marginal probabilities from logit or probit models. The coefficients from LPM are more easily understood and that is reason good enough.

However, what worries some of the row comparisons in Table 2 are misleading. People must have reported one of the four outcomes, say, for social distancing. Authors write that they “created a mutually exclusive set of indicator variables.” But they are interdependent, if a respondent chooses one option, then the other ones are excluded by design. If that is the case, one cannot run separate analyses for each outcome, if I understand correctly. If the outcome is something like a Likert scale with specific ordering (none > sometime > often > always), there are statistical or econometric models to analyze such outcomes (ordered probit?). Authors must consider that.

6. I am glad that the authors have looked at the spill-over (p.17). How about confining the samples to counties surrounding the “intervention” counties or using distance from those counties an additional “dose” variable?

Reviewer #1: No

Reviewer #2: No

---

## [Author Response · Author response to Decision Letter 0]

7 Sep 2020

REVIEWER 1

1. The authors present difference-in-difference estimates of the effect of the shelter-in-place (herein "lockdown") orders issued by seven U.S. counties around and including San Francisco, California, on survey respondents’ concerns and self-reported social distancing behavior. In their DID analysis, they find that Bay Area respondents are more likely to report social distancing “all of the time” (save for the youngest respondents, who actually social distance less than non-Bay-Area youth), and that Bay Area respondents post-lockdown had a harder time finding food relative, but otherwise had statistically insignificantly different challenges than non-Bay-Area respondents post-lockdown. They emphasize a number of limitations at the end of their study, all of which point towards their findings being an underestimate of the effect of lockdowns on individuals’ attitudes and behaviors. I find this paper interesting, but not as well executed as it should be. My two main concerns are the sampling technique (and the subsequent sample it provided, which the authors point out without dissembling) and the definition of their difference-in-difference.

We thank the reviewer one for their comments and their interest in the focus of our study. We have attempted to address the specific limitations of our study as detailed below. 

2. First, the sampling methodology: the authors use snowball sampling on three social media platforms over a period of just over two weeks. They end up with many thousands of responses who are deeply unrepresentative of the United States. To their credit, they neither paper over this fact nor neglect to mention it, but make it plain for the reader. And I agree with them that their sample (especially the educational background of their respondents) means that they likely underestimate the impact of the lockdown on the challenges people faced from sheltering in place. That is precisely why the authors should have chosen a different sampling approach from the get-go. That said, there’s nothing the authors can do about this now, and they have acknowledged it as much as they can.

We agree with Reviewer 1 that a more robust sampling approach (i.e. random digit dialing) would have facilitated collection of a more representative sample of responses. As the reviewer notes, we have attempted to be forthright about the limitations of the data and state in the methods section that this is a “cross-sectional, online survey with convenience sampling.” (Page 4, Line 3) We have further clarified in the “Results” section of the Abstract that this is a non-representative sample.

As the reviewer notes, we anticipate that the net effect of our sampling approach is to bias results toward the null and underestimate – rather than overstate – the impact of the shelter in place orders, but we also state in the limitations section that “difference in key demographic characteristics preclude generalization of our findings” (Page 10, Lines 12 – 14)

3. My second concern is something the authors can do something about. The authors analyze their data as if there is one and only one lockdown in all of the U.S.: the seven Bay Area counties start sheltering in place on March 16th. To put it bluntly: WHAT? Just three days later, *all of California* goes into lockdown. One week later, 40 percent of the U.S. by population is on lockdown, and just eleven days after the Bay Area begins sheltering in place, half of the United States shelters in place alongside it. [See usatoday.com/story/news/nation/2020/03/30/coronavirus-stay-home-shelter-in-place-orders-by-state/5092413002 for the list of lockdowns — I’m using the 2019 Census estimates of state population, and only including whole state lockdowns, which leaves out, e.g., both the major cities and nearly all the population of Pennsylvania.]

It seems to me that this is a pretty substantial oversight, thought as with the sampling approach, is one that leads the authors to understate potential differences between locked-down and non-locked-down individuals (as it treats many of the lockdown-treated as untreated). It also seems to me that the authors should instead be using something like a staggered difference-in-difference approach, given how many of their respondents likely (a) come from California and (b) come from other locked down states (because of the snowball sampling approach).

We agree with Reviewer 1 that the phased implementation of shelter-in-place announcements in California and throughout the U.S. in late March and early April of 2020 naturally suggests a staggered differences-in-differences approach, which the co-authors discussed extensively when were initially planning our analytic approach. Ultimately, the decision to conduct a difference-in-difference analysis focused only on the announcement in the Bay Area was made based on the fact that over 90% of survey responses were collected by March 19, 2020. This would make estimation of the effect of shelter-in-place orders outside of the Bay Area extremely imprecise, especially given the non-representative nature of the sample. 

We have included the following text in the methods section to illustrate this more clearly: “Because the majority of survey responses were collected by March 19, 2020, the DID analysis was focused on examining the impact of the Bay Area shelter-in-place announcement on March 16, 2020. The estimator compared the change in responses after versus before March 16, 2020 among respondents in the Bay Area versus elsewhere in the U.S.” (Page 6, Lines 9 – 12). We have also incorporated an additional figure (shown below) to the appendix (Supplemental Figure 1). This figure shows the daily (top panel) and cumulative (middle panel) number of responses over time. The figure also includes data from the article linked by Reviewer 1 (bottom panel). Ultimately, we feel that the highly imbalanced nature of the sample means that a staggered difference-in-difference approach would yield less precise estimates while necessitating stronger assumptions (e.g. that there was no growing concern as the number of state orders increased or concerns only changes with local state orders). We feel that the modeling approach presented in our manuscript is the most conservative approach and necessitates relatively fewer assumptions. We have incorporated a paragraph discussing this important limitation and further motivating our analytic decisions on Page 12 (Lines 12 – 19). 

Finally, given concerns raised by Reviewer 1 regarding the timing of shelter-in-place orders elsewhere and the number of respondents residing in California and other locked down states, we conducted several sensitivity analyses, which are presented in the appendix and largely consistent with the primary results:

• We excluded all respondents after March 19th, 2020. This sensitivity analysis removes the potential bias of other state-wide shelter-in-place orders. These results are shown in the appendix (Supplemental Table 3), discussed on Page 10 (Line 3) and are consistent with the primary results presented in the paper.

• We compared results for all California respondents versus all other US states, shown in the appendix (Supplemental Table 4). This sensitivity analysis relaxes the assumption that the Bay Area announcement did not affect other Californians (before the 3/19 state-wide announcement). On Page 6, Lines 25 – Page 7, Line 1 we state, “[b]ecause the announcement was highly publicized on mainstream news media channels and social media platforms, survey respondents living in California outside of the seven Bay Area counties may have modified their behaviors.”

• We compared results for the seven Bay Area counties and the state of Washington, which made another similar announcement on March 16th, 2020. (Supplemental Table 5)

4. In short: it is easy, four months later, to complain about an imperfect approach to getting respondents in the middle of a global pandemic. But inasmuch as PLOS One is less about getting exciting results and more about using good approaches (whatever results these may yield), both the sampling technique and the analysis do not meet that benchmark. The former is not fixable, but the latter is, and fixing it, I think, would improve the paper.

We thank the reviewer again for their comments and their trenchant critique. We believe, given the constraints of the sampling approach and distribution of survey responses, that our main analysis and sensitivity analyses combined with the figure and more thoughtful text make the paper’s conclusions more robust. 

REVIEWER 2

1. The paper, in my view, aims to share some very timely outcomes and analyses on impacts of COVID-19 in California, which was one of the first states to impose strict social distancing measure. The authors rightly zeroed on the seven counties who imposed the “shelter-in-place” policies. The authors looked at a number of outcomes using a quasi-experimental method (difference-in-difference, p.11). The DID estimates are somewhat modest with the largest impacts on increased difficulties with procuring food and mobility (transportation, p.14, also Table A2 and Figure 1). This may have important implications for lockdown policies in the US and elsewhere.

We thank Reviewer 2 for their assessment of our manuscript and address specific concerns below.

2. As a reader, I struggled to understand what the focus of the paper is, in terms of outcomes. The implications and focus of the title, the abstract and the introduction are not specific. The authors should consider what outcomes they are interested in and why.

We agree with Reviewer 2 that the choice of outcome measures could be better motivated. The three outcomes included in our analyses were intended to be sensitive enough to capture changes in behaviors and attitudes in response to COVID-19 at this early point in the natural history of the pandemic, but still represent meaningful impacts on individuals’ day-to-day experiences. The social distancing outcome was intended to capture early behavior changes in response to policy announcements (i.e. did individuals social distance more after the shelter-in-place orders were announced?). We asked how a wide range of day-to-day activities changed in the wake of the shelter-in-place announcement to gauge the impact on day-to-day experiences. Finally, we gauged how levels of concerns regarding the pandemic changed in the wake of the first shelter-in-place announcement. We have included additional text in the methods section to motivate our choice of outcome measures. (Page 5, Lines 17 – 20)

3. The description of the method is confusing. It is definitely a cross-sectional in the sense the data was collected at the individual levels at one point in time. However, it seems the authors managed to synthesize a panel by aggregating responses before and after imposition of the shelter-in-place policies by counties or geographic locations. This process is not very well described.

We have clarified this process in the methods. Specifically, we have now added a new subsection titled “Respondent Locations” in which we describe the process of assigning locations to participants. (Page 5, Lines 4 – 10) The aggregation process is also clarified in the difference-in-difference section, “compared the change in responses after versus before March 16, 2020 among respondents in the Bay Area versus elsewhere in the U.S.” (Page 6, Lines 11 - 12)

4. Related to #2 [here #3] above, it is important to let the reader know timing of different events. When did the government announce the shelter-in-place policy? How much in advance did the researchers and the citizens know about the policy? This is important because people may have time to adjust their behaviors and that can mitigate some of the outcomes that we eventually see in the study, for example, small size impacts on the outcomes the authors are interested in (again see Figure 1).

The focus of our analysis is on the impact of the shelter-in-place announcement on the outcomes of interest in the Bay Area versus elsewhere. The announcement occurred on March 16, 2020 and preceded the implementation of the order by three days (the order took effect on March 19, 2020). We state this in the methods section on Page 4, Lines 20 – 24 and additionally include a figure in the supplemental materials that demonstrates the timing of the announcement and the implementation of the order vis a vis cumulative number of surveys completed (Supplemental Figure 1). 

5. On page 14 (the manuscript does not have page numbers!), authors report difficulty in obtaining food. However, we don’t get a sense of the mechanism here. As people felt difficulty in moving from one place to another (another outcome the authors report), it is possible there was disruption in supply chain. Some discussion on this would be useful.

We apologize and have added page numbers to assist the reviewers with identifying changes. We anticipate the difficulty in obtaining food is likely due to surges in demand. As noted in media reports, the supply chain remained intact, but demand increased substantially. The following text is now included in the discussion section: “Respondents were most likely to report difficulty obtaining food, with increases in difficulty obtaining food more pronounced in the Bay Area following the shelter-in-place announcement. Difficulties obtaining food are most likely due to increased demand from consumers (rather than supply-side issues) as suggested by various media reports.” (Page 11, Lines 5 – 8) With citations for the following news reports incorporated:

https://www.nytimes.com/2020/03/15/business/coronavirus-food-shortages.html

https://www.npr.org/2020/03/18/817920400/empty-grocery-shelves-are-alarming-but-theyre-not-permanent

6. On page 11, LPM is all good. But I struggled understanding whether the same model can be applicable to all outcomes. Typically, they are applicable for binary outcomes and as robustness checks other models are used such as marginal probabilities from logit or probit models. The coefficients from LPM are more easily understood and that is reason good enough.

The initial rationale for using LPM is – as Reviewer 2 points out – they are easily interpreted. However, we agree that marginal probabilities from logit and probit models should be presented as robustness checks. We have done so (Page 7, Lines 4 – 7 in the methods section) and find that the overall pattern of findings with logit and probit models is largely consistent with those of the LPM presented in our main analysis. (Page 10, Lines 6 – 8 in the discussion section). The results of these robustness checks are presented in the Appendix in Supplemental Tables 6 – 8. 

7. However, what worries some of the row comparisons in Table 2 are misleading. People must have reported one of the four outcomes, say, for social distancing. Authors write that they “created a mutually exclusive set of indicator variables.” But they are interdependent, if a respondent chooses one option, then the other ones are excluded by design. If that is the case, one cannot run separate analyses for each outcome, if I understand correctly. If the outcome is something like a Likert scale with specific ordering (none > sometime > often > always), there are statistical or econometric models to analyze such outcomes (ordered probit?). Authors must consider that.

We apologize for the confusion regarding Table 2. It was not our intention to imply that ordinal responses were independent by using linear probability models. Our intention was to provide a descriptive summary in changes in each response level before and after the shelter-in-place orders with associated confidence intervals. We have revised our analysis and have updated Table 2 with estimates for percent changes with associated 95% CI calculated more appropriately with Yates’ corrected test of proportions. (Page 6, Line 2 and Table 2)

8. I am glad that the authors have looked at the spill-over (p.17). How about confining the samples to counties surrounding the “intervention” counties or using distance from those counties an additional “dose” variable?

We are concerned that restricting to counties surrounding the seven affected Bay Area Counties would lead to a substantial reduction in precision due to decreased sample size as only 1,874 respondents were residing in California but outside of the seven affected counties when they completed the survey. We also anticipate incorporating a distance metric into the analysis would require strong assumptions (i.e. that distance from the seven affected counties was a proxy for the impact of the announcement) that cannot be tested empirically and are unlikely to be satisfied given how widely the shelter-in-place announcement was publicized on local and national media. We present a number of sensitivity analyses that exclude respondents from Washington and California, respectively, to evaluate spillover effects (Page 10, Lines 2 – 8 and Supplemental Tables 3 - 5)

---

## [Decision Letter · Decision Letter 1]

3 Nov 2020

PONE-D-20-20290R1

The Impact of the first COVID-19 shelter-in-place announcement on social distancing, difficulty in daily activities, and levels of concern in the San Francisco Bay Area: A cross-sectional social media survey

PLOS ONE

Dear Dr. Linos,

Thank you for submitting your manuscript to PLOS ONE. After careful consideration, we feel that it has merit. Both referees are pleased with your revisions. However, I have two comments which should be addressed before the paper is accepted for publication. Therefore, we invite you to submit a revised version of the manuscript. My concerns are listed below.

1. The abstract says that “There is limited empirical research that examines the impact of these orders” but this claim is not substantiated. While this is true that there's a lack of empirical research, the evidence base is growing quite fast. For the US, there’re a number of COVID-19 impact studies conducted by economists on behavioral responses and socio-economic outcomes. But I don’t see any reference to that. As it stands, the paper doesn’t engage with the wider academic literature to support the claim made in the abstract. The reference list is rather lop-sided as the entire social science quantitative literature on COVID-19 is ignored. A quick search of the NBER database indicates nearly 50 articles: https://www.nber.org/search?page=1&perPage=50&q=covid-19

Some of these studies also employ DID estimators (e.g. Gupta et al (2020) Tracking Public and Private Responses to the COVID-19 Epidemic: Evidence from State and Local Government Actions)

I suggest that the authors cite some of these studies, acknowledge the different methodological approaches employed in the emerging literature on COVID-19 and in tat context identify the existing applications of DID to study the impact of COVID-19 in the US. I suggest that, based on the above, the authors update the “discussion section” and also add a para in “section 1” and the “concluding section” clarifying how their findings have added to the emerging evidence on the issue using US data.

2. Reviewer 1 suggests that the authors reorganize the paper around the supplemental analysis making it the main focus. I don’t think it’s necessary but I suggest that the supplemental tables are retained in the main body instead of being presented in the appendix. That way, they’ll receive equal attention from the readers.

In the next round, the revised version will not be sent out for external review. So kindly summarize in your cover letter how you have revised in response to my comments. A marked-up copy of your manuscript that highlights changes made to the original version. You should upload this as a separate file labeled 'Revised Manuscript with Track Changes'.An unmarked version of your revised paper without tracked changes. You should upload this as a separate file labeled 'Manuscript'.

We look forward to receiving your revised manuscript.

Kind regards,

M Niaz Asadullah

Academic Editor

PLOS ONE

Reviewers' comments:

Reviewer's Responses to Questions

**Comments to the Author**

1. If the authors have adequately addressed your comments raised in a previous round of review and you feel that this manuscript is now acceptable for publication, you may indicate that here to bypass the “Comments to the Author” section, enter your conflict of interest statement in the “Confidential to Editor” section, and submit your "Accept" recommendation.

Reviewer #1: All comments have been addressed

Reviewer #2: All comments have been addressed

2. Is the manuscript technically sound, and do the data support the conclusions?

Reviewer #1: Yes

Reviewer #2: Yes

3. Has the statistical analysis been performed appropriately and rigorously? 

Reviewer #1: Yes

Reviewer #2: Yes

4. Have the authors made all data underlying the findings in their manuscript fully available?

Reviewer #1: Yes

Reviewer #2: No

5. Is the manuscript presented in an intelligible fashion and written in standard English?

Reviewer #1: Yes

Reviewer #2: Yes

6. Review Comments to the Author

Reviewer #1: Thank you. You have addressed my concerns as well as anyone possibly could, and I appreciate it. As a matter of taste, I would make the supplemental analysis the main analysis, and show that things are more strongly against you if you use all the data and your original specification, but your choice is just as reasonable. Thank you for the supplemental tables.

Reviewer #2: I do not have any further comments on the revised version of the manuscript. Authors have sufficiently addressed the issues I raised earlier. I want to thank you the authors for accommodating the comments.

7. PLOS authors have the option to publish the peer review history of their article (what does this mean?). If published, this will include your full peer review and any attached files.

Reviewer #1: No

Reviewer #2: No

---

## [Author Response · Author response to Decision Letter 1]

15 Dec 2020

COMMENT 1: Thank you for updating your data availability statement. You note that your data are available within the Supporting Information files, but no such files have been included with your submission. At this time we ask that you please upload your minimal data set as a Supporting Information file, or to a public repository such as Figshare or Dryad.

Please also ensure that when you upload your file you include separate captions for your supplementary files at the end of your manuscript. As soon as you confirm the location of the data underlying your findings, we will be able to proceed with the review of your submission.

RESPONSE: In response to this comment, we have consulted with our Institutional Review Board and have reviewed our original IRB protocol. We have confirmed that – based on our original IRB protocol – we cannot make the de-identified dataset available with the manuscript due to concerns for re-identification given the fairly detailed nature of the survey. We can however make a de-identified dataset available to other researchers upon reasonable request. We have modified our data availability statement accordingly. We hope this alternative arrangement will be acceptable. 

COMMENT 2: Please ensure that you refer to Tables 6-10 in your text as, if accepted, production will need this reference to link the reader to the Tables.

RESPONSE: Our manuscript references Tables 5-7 and Tables 6-10 in the results section under the subheading “Sensitivity Analyses” on Page 10 (Lines 11-16).

---

## [Editor Report · Decision Letter 2]

17 Dec 2020

The Impact of the first COVID-19 shelter-in-place announcement on social distancing, difficulty in daily activities, and levels of concern in the San Francisco Bay Area: A cross-sectional social media survey

PONE-D-20-20290R2

Dear Dr. Linos,

We’re pleased to inform you that your manuscript has been judged scientifically suitable for publication and will be formally accepted for publication once it meets all outstanding technical requirements.

Kind regards,

M Niaz Asadullah

Academic Editor

PLOS ONE
---

## [Editor Report · Acceptance letter]

4 Jan 2021

PONE-D-20-20290R2 

The Impact of the first COVID-19 shelter-in-place announcement on social distancing, difficulty in daily activities, and levels of concern in the San Francisco Bay Area: A cross-sectional social media survey 

Dear Dr. Linos:

I'm pleased to inform you that your manuscript has been deemed suitable for publication in PLOS ONE. Congratulations! Your manuscript is now with our production department. 

Kind regards, 

on behalf of

Dr. M Niaz Asadullah 

Academic Editor

PLOS ONE